# Optical Genome Mapping in Routine Cytogenetic Diagnosis of Acute Leukemia

**DOI:** 10.3390/cancers15072131

**Published:** 2023-04-03

**Authors:** Gwendoline Soler, Zangbéwendé Guy Ouedraogo, Carole Goumy, Benjamin Lebecque, Gaspar Aspas Requena, Aurélie Ravinet, Justyna Kanold, Lauren Véronèse, Andrei Tchirkov

**Affiliations:** 1Cytogénétique Médicale, CHU Clermont-Ferrand, CHU Estaing, 63000 Clermont-Ferrand, France; 2Service de Biochimie et Génétique Moléculaire, CHU Clermont-Ferrand, 63000 Clermont-Ferrand, France; 3CNRS, INSERM, iGReD, Université Clermont Auvergne, 63001 Clermont-Ferrand, France; 4INSERM U1240 Imagerie Moléculaire et Stratégies Théranostiques, Université Clermont Auvergne, 63000 Clermont-Ferrand, France; 5Hématologie Biologique, CHU Estaing, 63100 Clermont-Ferrand, France; 6Hématologie Clinique Adulte et de Thérapie Cellulaire, CHU Estaing, 63100 Clermont-Ferrand, France; 7Service d’Hématologie et d’Oncologie Pédiatrique et Unité CRECHE (Centre de REcherche Clinique CHez l’Enfant), CHU Estaing, 63100 Clermont-Ferrand, France; 8Clonal Heterogeneity and Leukemic Environment in Therapy Resistance of Chronic Leukemias (CHELTER), EA7453, Université Clermont Auvergne, 63000 Clermont-Ferrand, France

**Keywords:** optical genome mapping, cytogenetics, acute leukemia, routine diagnostic procedures

## Abstract

**Simple Summary:**

Acute leukemia is a rare but severe disease for which genetic characterization allows diagnostic and prognostic classification and therapeutic management. The aim of this prospective study was to evaluate the performance of optical genome mapping (OGM) in providing an accurate genetic description of acute leukemia. We used OGM to analyze 29 samples of all types of acute leukemia collected at the time of diagnosis and compared the results obtained from karyotype, FISH, and RT-PCR, the standard techniques of our routine procedure. Overall, the results of OGM and standard techniques were concordant in all types of acute leukemia and lead to similar patient risk stratification. In addition, OGM confirmed its better resolution and detected additional genetic alterations of clinical interest. We conclude that OGM could effectively replace standard techniques and simplify our workflow in the diagnosis of acute leukemia.

**Abstract:**

Cytogenetic aberrations are found in 65% of adults and 75% of children with acute leukemia. Specific aberrations are used as markers for the prognostic stratification of patients. The current standard cytogenetic procedure for acute leukemias is karyotyping in combination with FISH and RT-PCR. Optical genome mapping (OGM) is a new technology providing a precise identification of chromosomal abnormalities in a single approach. In our prospective study, the results obtained using OGM and standard techniques were compared in 29 cases of acute myeloid (AML) or lymphoblastic leukemia (ALL). OGM detected 73% (53/73) of abnormalities identified by standard methods. In AML cases, two single clones and three subclones were missed by OGM, but the assignment of patients to cytogenetic risk groups was concordant in all patients. OGM identified additional abnormalities in six cases, including one cryptic structural variant of clinical interest and two subclones. In B-ALL cases, OGM correctly detected all relevant aberrations and revealed additional potentially targetable alterations. In T-ALL cases, OGM characterized a complex karyotype in one case and identified additional abnormalities in two others. In conclusion, OGM is an attractive alternative to current multiple cytogenetic testing in acute leukemia that simplifies the procedure and reduces costs.

## 1. Introduction

Acute forms of leukemia, such as acute myeloid leukemia (AML) and acute lymphoblastic leukemia (ALL), are hematologic malignancies resulting from aberrant differentiation and proliferation of transformed hematologic stem cells that lead to an accumulation of immature cells within the bone marrow and the suppression of both the growth and differentiation of normal blood cells. These malignancies arise from progenitor cells that have acquired chromosomal aberrations or somatic mutations conferring selective advantage. Up to 65% of acute leukemia cases in adults and 75% in children are diagnosed with chromosomal aberrations [1]. Genetic profiling is required to detect underlying abnormalities that define disease categories [2,3] and to establish risk prognostication [4] and therapeutic management. The current cytogenetic workflow in acute leukemia consists of mandatory karyotype, usually combined with targeted fluorescent in situ hybridization (FISH), and/or reverse transcriptase (RT)-PCR for detection of aberrant gene fusions [5,6,7]. The broad spectrum of genetic markers and some karyotype failures lead to the use of multiple FISH probes and RT-PCR assays. The multiplication of techniques and assays is time-consuming and costly.

Optical genome mapping (OGM) is emerging as an advantageous candidate to replace the gold standard multi-step procedure for leukemia diagnosis. The technique is based on a single enzyme labelling of ultra-high molecular weight (UHMW) DNA molecules that yield a subsequent high-resolution reconstruction of the patient’s genome [8]. Thus, it provides a whole genome analysis and can detect copy number variations (CNVs), and balanced and unbalanced structural variations (SVs). Early studies showed the potential clinical interest of OGM, and pilot studies have proposed approaches for adopting OGM in a clinical setting. OGM has been evaluated for the molecular characterization of AML [9,10] and ALL [11]. Using a cohort of 52 selected cases of diverse hematologic malignancies, Neveling et al. compared OGM with standard-of-care tests and observed a high analytical validity of OGM that gave a sensitivity of 100% and a positive predictive value of >80% [9]. In a series of 41 selected cases of ALL, OGM showed a good performance rate by correctly diagnosing 32/33 patients [12]. In a larger cohort of 100 randomly selected AML patients, OGM detected all clinically relevant SV and CNV and additionally supplied clinically relevant information in 13% of cases that had been undetected by the routine methods [13]. However, adopting OGM for routine cytogenetic diagnosis in acute leukemia needs further evaluation and validation in prospective studies. We report here a prospective study that aimed (i) to evaluate the use of OGM in routine diagnosis of acute leukemia cases in our laboratory, and (ii) to assess the relevance of OGM as an alternative approach to the standard-of-care tests combining karyotype, FISH and RT-PCR.

## 2. Materials and Methods

### 2.1. Sample Selection

Bone marrow (BM) aspirates (heparin or EDTA) or peripheral blood (PB) samples were collected from a series of cases sent to the Cytogenetics Laboratory of the University Hospital of Clermont-Ferrand, France, in the context of suspected acute leukemia. All patients underwent BM aspiration to ascertain the diagnosis of acute leukemia according to the World Health Organization (WHO) and the French-American-British (FAB) classifications.

Sampling was performed in a prospective manner from October 2021 to May 2022 in patients with sufficient left-over material for additional OGM analysis after routine testing. PB samples or BM aspirates were anonymized and frozen for further OGM procedures. Patients who had been previously treated for acute leukemia were excluded. All patients gave written consent for the use of results for research purposes and scientific publication.

### 2.2. Cytogenetics and Molecular Analyzes

The standard procedure followed the national recommendations [5,6]. Chromosome banding analysis (CBA) with G-banding was performed on chromosome preparations obtained after cell culture for 24 and/or 48 h without stimulation, and at least 20 metaphases were analyzed in case of the absence of chromosomal aberrations. Karyotypes were described according to ISCN 2020 [14]. Appropriate complementary FISH analyzes were performed in the same chromosome preparations with commercially available probes according to the manufacturer’s protocol (Abbott Molecular, Abbott Park, IL, USA, and MetaSystems, AmpliTech SAS, Compiègne, France). Molecular studies by RT-PCR to detect relevant fusion transcripts were performed on an uncultured sample fraction [15,16]. AML patients were stratified according to ELN recommendations [4].

### 2.3. Ultra-High Molecular Weight DNA Extraction and Labeling

The OGM procedure was performed at the Bionano Services Lab on the Gentyane Plateforme in Clermont-Ferrand. Samples for OGM were anonymized and stored at −80 °C within one day after collection. For most BM aspirates, aliquots were stored after washing with DNA stabilizing buffer, centrifugation, and removal of supernatant. PB samples and some BM aspirates were stored directly after the addition of DNA stabilizing buffer. Ultra-high molecular weight (UHMW) DNA was purified from a minimum of 1 mL of the blood sample, 1 mL of BM aspirate, or 1.5 million cells, following the manufacturer’s instructions (Bionano Genomics, San Diego, CA, USA). The following protocols were used: ‘Bionano Prep SP Frozen Human Blood DNA Isolation Protocol v2-30395, Rev B’ for the blood samples; ‘Bionano Prep SP Frozen Cell Pellet DNA Isolation Protocol v2-30398, Rev B’ for the BM cell pellets (both protocols from the SP Blood & Cell Culture DNA Isolation Kit); and the ‘Bionano Prep SP BM aspirate DNA Isolation Protocol v2-30399, Rev B’ for the BM aspirate (from the SP BM aspirate DNA Isolation Kit). Prior to DNA isolation, frozen samples were thawed in a 37 °C water bath, and white blood cells were counted and pelleted. For the samples with BM cell pellets, cells were washed twice with DNA stabilizing solution. Briefly, for the three sample types, cells were treated with proteinase K and LBB lysis buffer to release genomic DNA (gDNA). After the inactivation of proteinase K by PMSF treatment, gDNA was bound to a paramagnetic disk, washed, and eluted in an appropriate buffer. UHMW gDNA was incubated overnight at room temperature for homogenization. The following day, DNA molecules were labeled using the ‘Bionano Prep Direct Label and Stain (DLS) Protocol-30206, Rev G′ (Bionano Genomics, San Diego, CA, USA). Briefly, 750 ng of gDNA were labeled in the presence of Direct Label Enzyme (DLE 1) and DL-green fluorophores. The excess of DL-Green fluorophores was cleaned-up and the remaining DLE-1 enzyme was rapidly digested by proteinase K. Later, gDNA was counterstained overnight to visualize the DNA backbone.

### 2.4. Data Collection, Rare Variant Analysis, Structural Variant Calling, and Filtering

Labeled gDNA solution at a concentration between 4 and 12 ng/µL was loaded on a Saphyr chip and imaged by the Saphyr instrument (Bionano Genomics, San Diego, CA, USA). The Saphyr chip was run at a target of 1500 Gbp aiming at 400× coverage. For each sample, effective coverage of a minimum of 300× was achieved, enabling a theoretical mean variant frequency (VAF) sensitivity of 5%. Standard quality control parameters were assessed according to the manufacturer’s instructions and consisted of the total DNA collected ≥150 kb, the map rate, the N50 (≥150 kb), the average label density (in labels/100 kb), and the positive and negative label variance. Data analysis was performed in a single-blinded fashion using rare variant analysis. The rare variant pipeline (RVP) and variant calling were executed on Bionano Solve software (v3.7). Reporting and direct visualization of structural variants were made on Bionano Access (v1.7.1), using Genome Reference Consortium GRCh38/hg38 as the reference. SV and CNV were identified based on discrepancies in the alignment between the sample and the reference. For each SV and CNV call, confidence scores were calculated and provided by Bionano Genomics [17]. For data filtering, the following confidence scores were applied: insertion/deletion = 0; inversion = 0.7; duplication = −1; translocation = 0.05; copy number = 0.99; aneuploidy = 0.95. Filter settings were set to detect all CNV ≥ 10 kb and SV > 5000 bp. The self-molecule count was set at 5. All variants present in an OGM dataset of human control samples [18] from Bionano genomics were filtered out.

## 3. Results

### 3.1. Patient Characteristics

A total of 29 samples of acute leukemia were analyzed. They comprised 25 BM aspirates and four PB samples. The median patient age at diagnosis was 61 years (range 8–84). The male–female ratio was 1.31. According to the WHO classification of hematolymphoid tumors [2,3], two patients were diagnosed with B-cell acute lymphoblastic leukemia/lymphoma ALL (B-ALL), three patients with T-cell acute lymphoblastic leukemia/lymphoma (T-ALL), and 24 with acute myeloid leukemia (AML). The AML cases were classified into ELN 2022 risk categories by cytogenetic and mutational profiles (Appendix A). Table 1 shows the detailed patient characteristics.

### 3.2. OGM Quality Report and Results

For OGM analysis, effective coverage of >300× (mean = 474×, S.D. = 78.51) was achieved for all samples, with an average label density of 15.18/100 kb (S.D. = 0.43), map rate of 90.94% (S.D. = 6.2) and N50 (≥150 kb) of 296.4 kbp (S.D. = 27.1) (Appendix A). Sample #04 had a map rate value of 62.5%, below the recommended threshold of 70%. However other quality parameters were within the recommended range. This lower map rate was compensated by a slightly higher throughput to reach an effective coverage of 352×, and this sample was further analyzed without difficulties. After filtering, 1141 SV and 340 CNV were detected by OGM. Before analysis, we removed SV present in duplicates (*n* = 199) and translocations with a confidence score < 0.3. We also retrieved the CNV that matched an identical duplication and deletion identified by the SV pipeline (*n* = 34). We further merged the CNV describing the same rearrangement and removed CNV sizing less than 500 kb. Finally, a total of 785 SV and 71 CNV were considered for analysis (Appendix A) and comparison with results from routine techniques. Each SV was the subject of manual inspection to determine clinical relevance. To evaluate the concordance between CBA and OGM, only SV or CNV involving chromosomal segments of ≥10 Mb were considered to be detectable on the karyotype.

### 3.3. Comparison between OGM and Standard Procedure Results

Based on the results of the diagnosis procedure, the samples were classified into different categories (Figure 1): 8 samples (6 AML, 2 T-ALL) with normal standard test results, meaning that they did not harbor any chromosomal aberration or gene fusion; 11 samples (10 AML, 1 B-ALL) with a limited number of defined genetic abnormalities, i.e., they harbored one or two abnormalities, including karyotyping abnormalities or specific aberrant transcripts listed in the WHO classification [3]; and the remaining 10 samples (8 AML, 1 B-ALL, 1 T-ALL) with complex karyotype defined by the presence of three or more genetic alterations. Figure 1, and Table 2 and Table 3 show the results of standard tests and OGM for each patient.

#### 3.3.1. Leukemia without Abnormalities by Standard Techniques

Standard techniques found no genetic abnormalities in six AML cases and two T-ALL cases. OGM results were concordant for five out of six AML cases and one T-ALL. Regarding discrepancies, OGM detected trisomy 13 with a VAF of 14% in one AML case (#10). Because the cytogenetic preparation from this case was no longer available, additional FISH analysis was performed on the fixed preparation obtained from the concomitant blood sample, which initially contained 74% blast cells. This FISH analysis (XA Aneuscore probes from Metasystems) confirmed the presence of trisomy 13 in 15% of nuclei (Appendix A). OGM also detected a 40 Mb deletion of the long arm of chromosome 6 (del(6)(q13q22.1)) in the T-ALL case #11. The 6q21 deletion was confirmed by FISH analysis (XL 6q21/6q23 probe, MetaSystems) in 14 metaphases/20 (Appendix A). This abnormality was not detected by the CBA probably because of poor resolution.

#### 3.3.2. Leukemia with One or Two Molecular/Genetic Abnormalities by Standard Techniques

Three AML cases were diagnosed with a single aneuploidy (trisomy 8, trisomy 22, or loss of chromosome Y) after CBA, with similar results by OGM.

One AML case (#16) harbored an isolated translocation t(9;22)(q34;q11) with *BCR::ABL1* fusion. One case of acute promyelocytic leukemia (#09) harbored the characteristic translocation t(15;17) with *PML::RARA* fusion. Both translocations were also shown by OGM.

In AML case #25, OGM clarified the boundaries of the interstitial deletion of the long arm of chromosome 9 observed on the karyotype and confirmed the inversion of chromosome 14 as a fusion of region 14q13.2 with region 14q32.3. In AML case #01, OGM findings were also concordant with those of cytogenetic techniques, which described an extra chromosome generated by complex rearrangement of chromosome 8q (der(?)(?cen::8q24.3→8q22.1::8q22.1→8qter)). Thus, OGM detected partial trisomy of region 8q24.3 and partial tetrasomy of 8q22.1–8q24.3 by the CNV pipeline associated with an SV corresponding to 8q22.1 duplication.

In contrast, in two AML cases, abnormalities reported by CBA were not detected by OGM. In AML case #19, the karyotype described an additional isochromosome of the long arm of chromosome 8 associated with a der(13;14)(q10;q10) similar to a Robertsonian translocation and potentially of constitutional origin. OGM partially confirmed the tetrasomy for the long arm of chromosome 8, revealing an interstitial submicroscopic deletion (2.31 Mb) at 8q24.3, but it missed translocation t(13;14). This discrepancy was expected as the detection of abnormalities in repeated regions is a known limitation of the OGM technique. In AML case #14, the cytogenetic analysis showed, in 6 out of 20 metaphases, a large deletion of the short arm of chromosome 7 from band 7p12. This deletion was associated with a minute chromosome made of chromosome 7 material as confirmed by chromosome painting (Figure 2). OGM software failed to detect any abnormality of chromosome 7, but visual inspection enabled observation of a fusion between 7p21.3 and 7p11.2, which might be a result of 7p deletion. Removal of SV filters allowed the software to detect this intra-chromosomal fusion with a good confidence score of 0.86. The 7p loss was not detected by the CNV pipeline, probably due to a low VAF (2%).

In two cases, OGM detected additional aberrations compared to the routine procedure. In AML case #06, routine testing showed a pericentric inversion of chromosome 16 with *CBFB::MYH11* fusion that was correctly detected by OGM (VAF = 47%) but OGM analysis also showed an additional subclone with loss of chromosome Y with VAF of 14%. In B-ALL case #18, karyotyping was normal. However molecular analysis detected an *ETV6::RUNX1* fusion transcript resulting from a cryptic translocation t(12;21). The presence of *ETV6::RUNX1* fusion was confirmed by FISH. OGM outperformed routine diagnosis since it detected the t(12;21) with VAF of 47% but also showed multiple additional aberrations including a chromothripsis event involving chromosomes X, 10, and 13 (Appendix A).

#### 3.3.3. Leukemia with Complex Karyotype

In ten cases (1 T-ALL, 1 B-ALL, 8 AML) routine CBA revealed a complex karyotype.

In four of these cases (B-ALL #03 and AML #12, #13, and #27), OGM data were considered completely concordant with some clarifications of the chromosomal breakpoints (Appendix A).

In three AML cases (cases #07, #15, #23) OGM analysis failed to detect additional clones or subclones. More specifically, in case #15, CBA showed a complex karyotype with a dicentric chromosome (5;10) and trisomy 21 associated with a subclone presenting several additional chromosome gains. OGM was able to describe the main clone with greater accuracy, unravelling a complex rearrangement between the two chromosomes 5 and chromosome 10 but missed the subclone. In AML case #07, karyotyping detected three apparently independent clones. OGM identified the clone with translocation t(3;8), confirming the rearrangement of the *MECOM* locus, but overlooked the other clones (Appendix A). CBA showed a translocation t(9;12), while OGM detected a minor clone with a translocation t(8;12) (VAF = 5%). The t(9;12) was confirmed by FISH using chromosome painting showing complex rearrangement of chromosome 9, a small derivative chromosome 12, and a derivative chromosome painted with probes of both chromosomes 9 and 12. Finally, the translocation t(8;12) detected by OGM was not validated and not considered in the final OGM result. OGM analysis also identified an intra-chromosomal fusion in the long arm of chromosome 5 between positions 87.4 Mb (5q14.3) and 144.5 Mb (5q31.3) suggesting a deletion. This low VAF (4%) deletion was not detected with the CNV pipeline. FISH analysis (performed with XL del(5)(q31) probe, MetaSystems) did not confirm any deletion or aberrant position of the *EGR1* locus (5q31.2) on 20 metaphases and 200 nuclei. Similarly, in AML case #23, OGM failed to detect a subclone containing an extra copy of deleted chromosome 9, which was observed in only 2 out of 20 metaphases.

In the remaining three cases with a complex karyotype (T-ALL #29, AML cases #02, and #04), OGM analysis provided a more detailed description of the abnormalities, enabling the chromosomal formula to be corrected. For instance, regarding case #02, OGM showed that (i) the detected monosomies of chromosomes 15 and 16 were only partial and not complete as concluded by CBA analysis, maybe related to the presence of a marker chromosome, (ii) the derivative chromosome der(7) also contained a partial deletion of the short arm, and (iii) there was no deletion on chromosome 9 (Appendix A). In case #29, OGM confirmed the translocation t(X;10) resulting in the *DDX3X::MLLT10* fusion and revealed a three-way translocation instead of the t(1;7) initially identified by CBA (Figure 3). More interestingly, OGM showed that this translocation interrupts the *TP73* gene between introns 1 and 3. This interruption could lead to the production, by an alternative promoter, of a *TP73* isoform that lacks the amino-terminal transactivation domain [19] and is able to promote oncogenesis through dominant negative effects on *TP53* and full-length *TP73* [20]. In addition, although OGM detected the gain of region 7q22.2q36.3 described in the karyotype as derivative chromosome 19 with insertion of 7q material, it failed to detect the localization of this gain on chromosome 19. However, visual inspection of the molecules detected a chimeric map that fused the 19p13.3 region with the limit of the CNV gain of chromosome 7q (Figure 3). The presence of repeated sequences in the telomeric region of chromosome 19 could explain why this SV was not called by the software.

### 3.4. Additional Abnormalities of Clinical Interest Detected by OGM

OGM detected additional abnormalities not described by standard techniques including aberrations cryptic at the karyotype level in several patients. First, in B-ALL case #03 with complex karyotype and *BCR::ABL1* fusion, OGM detected infrachromosomal deletions characteristic of B-ALL, namely recurrent deletions in *IKZF1*, *PAX5*, *BTG1*, *ADD3* genes, and *CDKN2A* biallelic deletion (Figure 4) [21]. OGM also showed a 135 kb-deletion encompassing the 5′-part of *KRAS* and a translocation t(12;13) inducing a putative novel fusion of the *WDFY2* gene with the *ARID2* gene, a component of the SWI/SNF complex with tumor suppressive properties that play a role in the differentiation of hematopoietic stem cells [22,23]. Interestingly this translocation is associated with an intragenic deletion of *ARID2*, which could result in haploinsufficiency.

Second, in both cases of T-ALL, OGM provided additional information. In case #05, OGM showed two different deletions (4.8 Mb and 1.2 Mb) on the short arm of chromosome 12 resulting in a biallelic deletion of the *ETV6* and *CDKN1B* genes and a homozygous deletion of 1.63 Mb on chromosome 9p encompassing the *CDKN2A/B* genes. In addition, OGM showed an interstitial deletion of 2.46 Mb between 9q34.11 and 9q34.13. This deletion creates the well-known fusion *SET::NUP214* for which OGM failed to identify the *SET* involvement (Figure 5). Indeed, OGM localized the breakpoint at 9q34.11 in the *WDR34* gene instead of *SET*. A manual review confirmed that this breakpoint actually occurs in *SET* and showed a misalignment of two labels consecutive to a tiny difference in their distance to the last label correctly aligned to the reference. Of note, manual inspection of this case also detected a deletion of the *TCR gamma* (*TRG*) locus (Appendix A). This deletion was filtered because of its presence in the control database, so it should be considered with caution. Likewise, in the second T-ALL case #11, OGM detected two overlapping deletions at 9p22.1p21.3 region encompassing the *CDKN2A/B* genes and a deletion at 14q11.2 encompassing the *TCR alpha* locus. Finally, in AML case #01, OGM showed a duplication inside the *KMT2A* gene corresponding to the well-known partial tandem duplication (*KMT2A-PTD*) that was not otherwise confirmed.

OGM confirmed its superiority in describing abnormalities especially at the breakpoints and in unraveling complexity. Thus, as previously noted, in case #25, OGM detailed the inversion of the long arm of chromosome 14 and showed that the breakpoints are located next to the *BRMS1L* gene at 14q32.13, a gene involved in breast and ovarian cancer, and next to the *TCL1A* locus at 14q13.2. Interestingly, *TCL1A* would be more likely expected to be involved in lymphoid neoplasms as it plays a role in lymphopoiesis [24]. OGM also detected an additional translocation t(5;19) with a VAF of 2%. However, the confidence score was low (0.33), the rearrangement complex with internal inversions, and the breakpoint on chromosome 5 was located in a gene-free region close to the centromeric region. This translocation was not validated by FISH and seems to be a false positive SV call.

In AML case #04, OGM showed a particularly highly complex karyotype (Appendix A). Among abnormalities, OGM confirmed a translocation t(7;14) involving the *CDK6* gene at 7q21.2 and the *BCL11B* gene at 14q32.3. While the *CDK6* gene is known to be a critical regulator of normal and leukemic stem cells [25], the involvement of *BCL11B* in myeloid neoplasms is unusual [26]. OGM also showed aberrations involving chromosomes 3 and 17. In particular, it detected a translocation t(3;17) that disrupts the *TP53* locus.

### 3.5. Comparison of Implemented Resources of the Standard Procedure with OGM

In addition to the performance of the OGM technique in detecting and describing large and small chromosomal abnormalities, we aimed to determine the organizational benefits of using this technology instead of the current procedure in our laboratory. An important prerequisite for the diagnosis of acute leukemia is a turnaround time of five to ten days allowing rapid care and treatment initiation [4,7]. On the one hand, our standard workflow provides targeted results on chromosomal rearrangements in two to five days by RT-PCR and/or FISH analysis and a genome-wide analysis with karyotype in four to seven days. On the other hand, the OGM technique takes around five to seven days, which is a few days longer than molecular or FISH studies but equivalent to the standard procedure for complete results with a superior resolution. Hence, we could envisage routinely using the technology in those indications. However, the limitations of the OGM technique have to be kept in mind, especially in genomic repeated regions.

Importantly, the diagnosis of these 29 cases of acute leukemia involved the performance of 33 CBA (some of which were conducted in duplicate cases of normal or unsuccessful karyotype), 112 FISH analyzes, and 64 RT-PCR assays. In comparison, OGM provides results with a single test. In this series, OGM demonstrated its performance in detecting all critical abnormalities for acute leukemia characterization. Thus, its routine implementation in our laboratory could simplify the standard procedure thereby avoiding a significant number of assays.

One of the major benefits of OGM is the lower number of cells required for the analysis. This is particularly important for the successful analysis of paucicellular samples. Currently, OGM requires about 1.5 million cells while CBA requires a minimum of 10 million cells. For RT-PCR assays, about 2 to 5 million cells are usually needed. Moreover, OGM eliminates the cell culture step, thus avoiding artifact aberrations, selection of minor clones, and failures due to unsuccessful leukemic cell proliferation.

## 4. Discussion

### 4.1. Impact of OGM Results on Patient Risk Stratification According to International Classifications

Overall, OGM automatically detected 73% (53/73) of abnormalities found by the standard techniques of karyotyping, FISH, and RT-PCR. Many of the events that were missed belong to one case (#07), with low-frequency subclones that are likely below the OGM limit of detection. Nonetheless, the non-detection of some abnormalities by OGM analysis in five cases (#07, #14, #15, #19, #23) did not change the prognostic classification. Thus, OGM discrepancies had no or limited impact even when the number of chromosomal abnormalities was lower than those defined by the karyotype. In cases #14 and #19, the missing aberrations were not associated with any particular outcome in AML and did not modify the stratification group of the patients. This was especially the case for Robertsonian translocation, which is an acknowledged limitation of OGM. Although acquired Robertsonian translocations are not rare in leukemia, we were unable to certify that its occurrence was not a constitutional abnormality in this patient, who died rapidly after diagnosis. Case #07 was assigned to the group with *MECOM* rearrangement, which is generally associated with an unfavorable outcome [27], while case #15 was still considered to be a complex karyotype and therefore assigned to the high-risk group according to the ELN classification. Finally, case #23, who had an undetected subclone, was assigned to the favorable risk group due to a translocation t(8;21) with *RUNX1::RUNX1T1* fusion irrespective of any additional aberration including deletion 9q [28]. Likewise, in two cases, OGM detected minor clones or subclones that did not affect the stratification of the AML. In case #10, trisomy 13 did not confer any additional prognostic value compared to the normal karyotype, and case #06 was assigned to a favorable risk group relative to the *CBFB::MYH11* fusion. Regarding B- and T-ALL, the additional information provided by OGM analysis did not lead to any fundamental change in the characterization of leukemia.

In conclusion, all of the 27 abnormalities that had a diagnostic relevance and/or prognostic impact were correctly detected by OGM, and additional abnormalities reported by OGM did not modify the risk group the patients were assigned to. Thus, OGM had 100% concordance with standard techniques for the classification of AML and ALL according to the WHO and the International Consensus Classification [25] and for AML stratification according to the ELN classification.

### 4.2. Confirmation of the Additional Value of OGM in the Diagnosis of Acute Leukemia

In addition to the accurate classification of AML and ALL, OGM identified other aberrations which are of clinical interest. As previously reported, OGM is a powerful tool for the characterization of ALL. In our study, OGM provided a very precise molecular description of a *BCR::ABL1*-positive B-ALL case (#03), detecting deletions in *IKZF1*, *PAX5*, *BTG1*, *ADD3*, *CDKN2A,* and *KRAS* genes. Although *BCR::ABL1* has a major effect on risk stratification and therapeutic decision, OGM data could influence patient management since *IKZF1* alterations were associated with treatment resistance to tyrosine kinase inhibitors [29,30]. In addition, OGM identified a translocation and a deletion affecting the *ARID2* gene. Of note, mutations in ARID family genes were recently identified as biomarkers for treatment with immune checkpoint inhibitors [31].

T-ALL biology is considerably characterized by genetic alterations such as point mutations or chromosomal rearrangements, which affect oncogenic drivers, but few have been identified as independent prognosis markers. In our T-ALL cases, OGM successfully detected cryptic abnormalities. In case #05, it identified deletion 9q, which creates the *SET::NUP214* fusion. This finding could have an effect on patient management since *SET::NUP214* was associated with a higher risk of relapse [32]. In addition, *SET::NUP214* could serve as a molecular marker for MRD monitoring by digital PCR with better sensitivity than flow cytometry. A non-biallelic *TRG* deletion found in the same case was previously associated with poor survival [33]. A large deletion of chromosome 6q found in case #11 could influence therapeutic management since this aberration was associated with a poor outcome [34].

This study also confirmed the additional contributions of OGM to AML diagnosis. For instance, in AML case #04, identification of the alteration of *BCL11B* together with *CDK6* is of interest since this molecular profile is more frequently observed in the early T-cell precursor (ETP)-ALL than in AML. Duffield et al. recently suggested that it could represent a molecularly defined group of acute leukemia [26]. It is noteworthy that this AML also harbored an internal tandem duplication in the juxtamembrane domain of the FMS-like tyrosine kinase 3 (*FLT3-ITD*) gene detected by routine techniques. This *FLT3-ITD* is known to increase *CDK6* expression. Together, these abnormalities reinforce the hypothesis of the therapeutic relevance of CDK4/6 inhibitors in this patient [35]. Another example is AML case #01, in which OGM detected the *KMT2A-PTD*. The finding is of particular interest because the *KMT2A* alteration is an adverse prognostic factor in AML [36]. It could be a genetic marker at the time of diagnosis and for molecular monitoring during treatment [37,38,39]. It is also a putative therapeutic target. Of note, routine testing detected an *FLT3-ITD* in this patient and recent advances have shown a significant synergistic anti-leukemic effect of menin-KMT2A inhibitors in combination with FLT3 inhibitors [39].

### 4.3. OGM as a Routine Tool

Comparison of OGM with standard-of-care methods in the diagnosis of acute leukemia in this study confirmed that OGM is a very efficient technique in detecting clinically relevant cytogenetic and molecular abnormalities. The higher resolution compared to karyotyping allows the detection of additional abnormalities not found by standard techniques. OGM provided a more accurate description of abnormalities and breakpoint identification at the gene level, identifying the involvement of candidate genes with a known or putative role in leukemogenesis or as therapeutic targets, such as *TP53*, *TCL1A*, *KMT2A*, *CDK6*, or *BCL11B*. As previously described, OGM outperformed classical techniques for the identification of complex abnormalities in cases with multiple intra-chromosomal rearrangements and chromothripsis. This enhanced capacity could be of particular interest in disease staging in myelodysplastic syndromes for example [40]. The greater degree of genomic complexity provided by OGM could make it necessary to devise a specific OGM-based classification and standards for the adoption of OGM in the diagnosis of acute leukemia.

This study also confirmed that OGM fails to automatically detect rearrangements in chromosomal regions with repeated sequences and abnormalities involving genomic regions uncovered by enzymatic labeling. Some of these missed events were found after a manual revision of the maps. Nevertheless, such limitations could lead to a higher proportion of false negative diagnoses or underdiagnoses which should be taken into account and estimated. OGM also failed to detect abnormalities involving small-size clones, which were known to be below the limit of detection of the technique, but in our series, this was not a major issue in the diagnosis of acute leukemia. In addition, although the VAF level can provide clues, OGM results do not determine whether detected abnormalities occur in the same clone or not. It could be challenging, therefore, to count the number of reported abnormalities in each clone as described in ISCN and to unravel the clonal hierarchy. These limitations could be of critical importance in some other hematological malignancies like chronic lymphocytic leukemia [41,42]. However, the relevance of OGM counting of abnormalities should be confirmed by further studies. The relevance of the small size SV which is detected in great numbers by OGM also requires fuller characterization.

This series of acute leukemia cases provides strong arguments to consider OGM as an effective technique to replace our routine protocol since its findings were strongly correlated with standard-of-care test results. Our standard procedure involves numerous technicians with various specialized skills, especially for the manual technique of CBA. OGM too requires well-trained and qualified staff, especially for DNA extraction, the handling of software, filter settings, and interpretation to achieve rapid turnaround and accurate reports. Replacing standard techniques with OGM would effectively reduce the number of routine assays. However, overall cost savings would be limited owing to the still high price of reagents and equipment.

## 5. Conclusions

This prospective study validated the use of OGM in the routine diagnosis of acute leukemia in our laboratory and confirmed the benefit of adopting OGM alternatively to standard-of-care testing associating karyotyping, FISH, and RT-PCR. OGM allowed accurate diagnosis and correct classification of acute leukemia cases as required by the WHO and ELN. OGM also provided additional results whose interest should be evaluated with the aim of establishing new recommendations for the implementation of OGM in the management of hematologic malignancies.

## Figures and Tables

**Figure 1 cancers-15-02131-f001:**
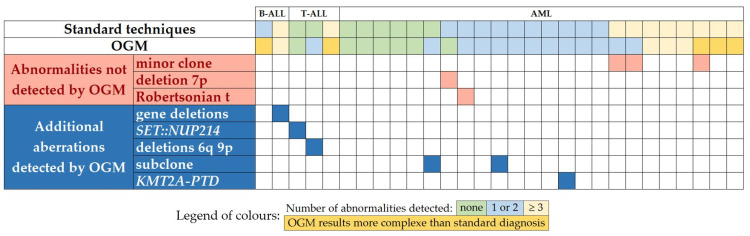
Comparison of routinely used standard techniques and OGM results.

**Figure 2 cancers-15-02131-f002:**
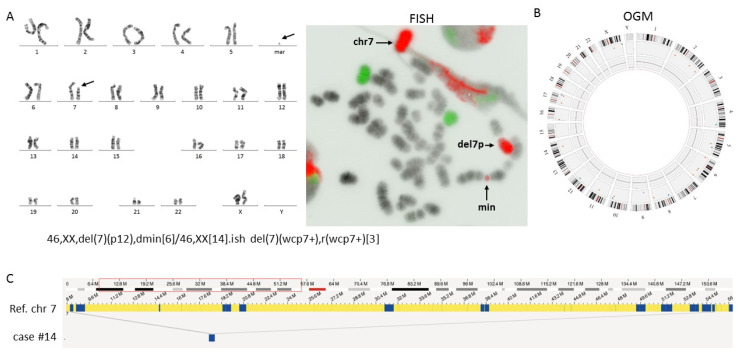
Cytogenetic and OGM results for case #14. (**A**) CBA showed a large deletion of the short arm of chromosome 7 associated with the presence of a minute chromosome; FISH performed with painting probes (MetaSystems) of chromosomes 7 (in red) and X (in green) showed that the minute chromosome was composed of chromosome 7 material. The karyotype is defined to reflect CBA and FISH results (×100). (**B**) The Circos plot obtained by OGM analysis showed no CNV or SV. (**C**) Visual inspection showed a genomic map connecting positions 8.44 Mb and 54.8 Mb on chromosome 7p. Ref. = reference genome, chr = chromosome.

**Figure 3 cancers-15-02131-f003:**
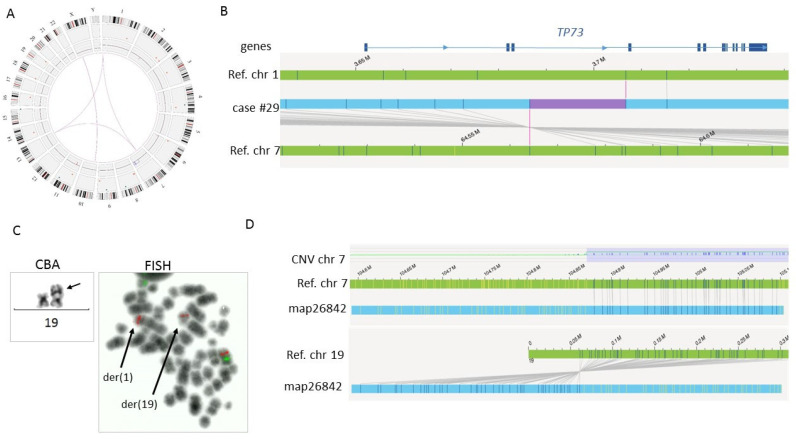
OGM results for case #29. (**A**) Circos plot showing t(X;10) and CNV gain of the long arm of chromosome 7 and showing that t(1;7) initially described on karyotype is actually a three-way translocation involving a chromosome 13. The translocation of the extra copy of chromosome 7q to chromosome 19 described by CBA and FISH was not automatically detected by OGM. (**B**) Schematic representation of the alignment of a genomic map corresponding to the translocation between chromosomes 1 and 7 showing that the breakpoint at chromosome 1 disrupts the *TP73* gene between introns 1 and 3 (blue box represents exons). (**C**) Detail of CBA (×60) showing the der(19) with additional material on the short arm. FISH (×100) with LSI D7S486 probe (Abbott) in red and control centromeric probe in green showed three spots for LSI D7S486: on the normal chromosome 7 with control probe, on der(1), and der(19). (**D**) Schematic representation of the alignment of genomic map 26,842 showing a match with the reference genome map of chromosome 7 at the limit of the CNV gain and with the reference genome map of chromosome 19 at the telomeric end of the short arm. CNV = copy number variation. der = derivative.

**Figure 4 cancers-15-02131-f004:**
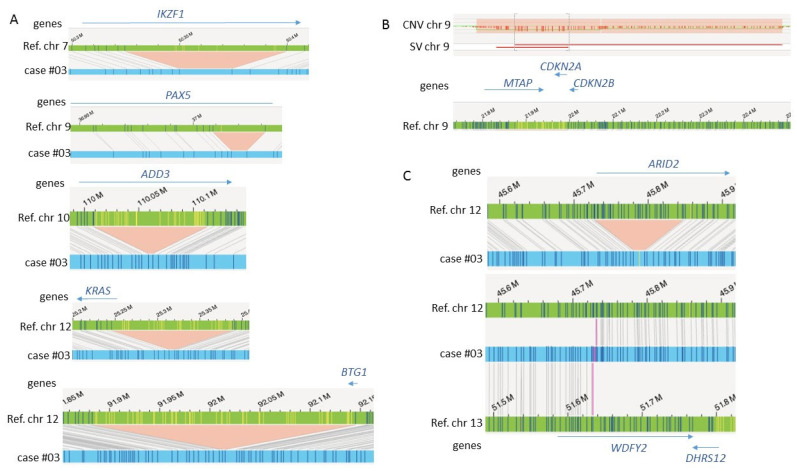
OGM results on case #03. (**A**) Schematic representations showing multiple cryptic deletions of genes of interest. (**B**) At region 9p21.3, the SV track detected two overlapping deletions represented by two different lines (in red) and resulting in a homozygous deletion region (in gray square brackets) inside larger heterozygous deletions. These deletions are visualized by two different levels of the line on the CNV track. (**C**) Schematic representation of the deletion detected by OGM in the *ARID2* gene ((**upper**) figure) and breakpoint of the t(12;13) localized next to the *ARID2* gene ((**lower**) figure).

**Figure 5 cancers-15-02131-f005:**
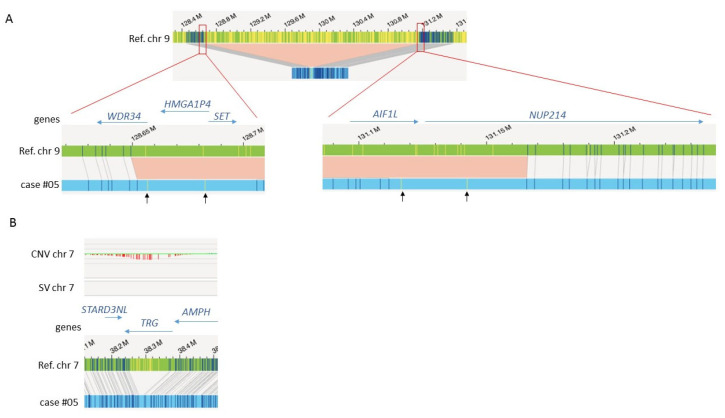
OGM results on case #05. (**A**) OGM detected a 2.46 Mb deletion at 9q34.11-q34.13 ((**upper**) figure) but did not identify the *SET::NUP214* fusion. Detailed views at the breakpoints ((**lower**) figures) depict two labels not aligned to the reference genome (indicated by black arrows) at any of the breakpoints so that the breakpoint at 9q34.11 is incorrectly located in the *WDR34* gene. If the two non-aligned labels are considered to align with the reference genome at 9q34.11, the breakpoint should be localized in the *SET* gene. (**B**) Deletion at the *TCR gamma* (*TRG*) locus that is not called by the SV or CNV pipelines, although the CNV track shows an inflection corresponding to the loss.

**Table 1 cancers-15-02131-t001:** Patient characteristics, source of sample and percentage of blasts, type of leukemia according to the WHO classification, subtype of AML according to the FAB classification, and risk category according to the ELN classification.

#ID	Age (Years)	Sex	Sample Type	Sample Blast Count (%)	Diagnosis *	AML FAB- Subtype	Risk Category
01	52	F	BM	97	AML	M1	Intermediate
02	70	F	BM	49	AML-MR	M1	Adverse
03	56	F	PB	70	B-ALL with *BCR::ABL1* fusion	n/a	n/a
04	19	M	BM	76	AML-MR	M4	Adverse
05	21	F	BM	89	T-ALL	n/a	n/a
06	74	M	BM	83	AML with *CBFB::MYH11* fusion	M4Eo	Favorable
07	63	M	BM	57	AML with *MECOM* rearrangement	sec. AML	Adverse
08	64	M	PB	51	AML	sec. AML	Intermediate
09	61	M	BM	87	APL with *PML::RARA* fusion	M3	n/a
10	49	F	BM	94	AML	M5	Adverse
11	14	M	BM	82	T-ALL	n/a	n/a
12	63	M	PB	86	AML with *KMT2A* rearrangement	M4	Adverse
13	84	M	BM	70	AML-MR	M5	Adverse
14	56	F	BM	62	AML	M1	Intermediate
15	61	F	BM	37	AML-MR	sec. AML	Adverse
16	64	M	BM	35	AML with *BCR::ABL1* fusion	sec. AML	Adverse
17	81	M	BM	79	AML	M0	Intermediate
18	8	F	BM	95	B-ALL with *ETV6::RUNX1* fusion	n/a	n/a
19	80	M	PB	79	AML	M4	Intermediate
20	23	F	BM	38	AML	M4	Intermediate
21	81	F	BM	82	AML	M1	Intermediate
22	60	M	BM	27	AML	M5	Favorable
23	34	F	BM	58	AML with *RUNX1::RUNX1T1* fusion	M2	Favorable
24	64	F	BM	76	AML	M4	Intermediate
25	66	F	BM	99	AML	M0	Intermediate
26	80	M	BM	88	AML	M4	Intermediate
27	60	M	BM	63	AML with *RUNX1::RUNX1T1* fusion	M2	Favorable
28	75	M	BM	87	AML	M4	Favorable
29	50	M	BM	100	T-ALL	n/a	n/a

#ID = sample number. * according to the WHO classification. M = male, F = female. AML-MR = AML myelodysplasia-related. FAB = French-American-British classification. sec. AML: secondary AML. BM = bone marrow. PB: peripheral blood cells. n/a: not applicable.

**Table 2 cancers-15-02131-t002:** Detailed results of standard testing versus OGM for 24 cases.

#ID	Leukemia Type	Standard Karyotype ^1^	FISH	Fusion Transcripts	Karyotype Description Based on OGM ^2^
18	B-ALL	46,XX[20]	ish t(12;21)(RUNX1+,ETV6+; RUNX1+,ETV6+)[6]	*ETV6:* *:RUNX1*	complex karyotype with t(12;21)
20	AML	46,XX[20]	nuc ish(KMT2A×2)[200],(CBFB×2)[200]		46,XX
21	AML	46,XX[28]	nuc ish(KMT2A×2)[200],(PML, RARA)×2[200],(CBFB×2)[200]		46,XX
22	AML	46,XY[22]	nuc ish(KMT2A×2)[200],(CBFB×2)[200]		46,XY
24	AML	46,XX[20]	nuc ish(KMT2A×2)[200],(CBFB×2)[200]		46,XX
28	AML	46,XY[20]	nuc ish(KMT2A×2)[200],(CBFB×2)[200]		46,XY
10	AML	46,XX[8] *	nuc ish(KMT2A×2)[200],(CBFB×2)[200]		47,XX,+13
14	AML	46,XX,del(7)(p12),dmin[6]/ 46,XX[15]	ish del(7)(wcp7+),r(7)(wcp7+)[3].nuc ish (KMT2A×2)[200],(CBFB×2)[200]		46,XX
19	AML	46,XY,+i(8)(q10),der(13;14) (q10;q10)?c[23]	nuc ish(KMT2A×2)[200],(CBFB×2)[200]		47,XY,+der(8)i(8)(q10) del(8)(q24.3q24.3)
06	AML	46,XY,inv(16)(p13q22)[18]/ 46,XY[2]	ish inv(16)(p13)(5′CBFB+) (q22)(3′CBFB+)[19].nuc ish(KMT2A×2) [200],(CBFB×2)(5′CBFB sep 3′CBFB×1) [66/100]	*CBFB:* *:MYH11*	46,XY,inv(16)(p13q22)/45,sl,-Y
08	AML	47,XY,+8[16]/46,XY[4]	nuc ish(KMT2Ax2)[200],(CBFBx2)[200]		47,XY,+8
09	AML	46,XY,t(15;17)(q24;q21)[17]/ 46,XY[3]	ish t(15;17)(PML+,RARA+;RARA+, PML+)[20].nuc ish(KMT2A×2)[200], (PML,RARA)×3(PML con RARA×2) [93/100]	*PML::RARA*	46,XY,t(15;17)(q24;q21)
16	AML	46,XY,t(9;22)(q34;q11)[20]	nuc ish(KMT2A×2)[200], (PML,RARA)×2[200],(CBFB×2)[200]	*BCR::ABL1*	46,XY,t(9;22)(q34;q11)
17	AML	47,XY,+22[11]/46,XY[10]	nuc ish(KMT2A×2)[200],(CBFB×2)[200]		47,XY,+22
25	AML	46,XX,del(9)(q13q22), inv(14)(q13q24)[20]	ish del(9)(ABL1+),22q11.2(BCR×2)[20]. nuc ish(ABL1,BCR)×2[200],(KMT2A×2) [198/200],(TCRA/D×2)[195/200], (CBFB×2)[198/200]		46,XX,del(9)(q21.11q31.1), inv(14)(q13.2q32.13)
26	AML	45,X,-Y[30]	nuc ish(KMT2A×2)[200],(CBFB×2)[200], (RARA×2)[50]		45,X,-Y
07	AML	40~45,X,-Y,add(7)(q21), t(9;12)(p?21;?q12),add(16) (q2?1),+mar,2dmin,inc[3]/ 46,XY,t(3;8)(q26;q21)[2]/47, XY,+der(1;19)(q10;p10)[1]/ 46,XY[21]	ish t(3;8)(3′MECOM+;5′MECOM+)[17]. nuc ish(MECOM×2)(3′MECOM sep 5′MECOM×1)[172/200]		46,XY,t(3;8)(q26;q21)
23	AML	46,XX,t(8;21)(q21;q22), del(9)(q13q22)[18]/47,sl, +del(9)(q13q22)[2]	nuc ish(KMT2A×2)[200],(CBFB×2)[200]	*RUNX1:* *:RUNX1T1*	46,XX,t(8;21)(q21;q22), del(9)(q21.11q31.1)
12	AML	47,XY,del(1)(q41q42),t(9;11) (p21;q23),+21[20]	ish t(9;11)(3′KMT2A+;5′KMT2A+)[40]. nuc ish(KMT2A×2)(5′KMT2A sep 3′KMT2A×1)[192/200],(CBFB×2)[200]	*KMT2A:* *:MLLT3*	47,XY,del(1)(q41q42),t(9;11) (p21;q23),+21[20]
13	AML	48,XY,del(1)(p36p34), +del(1)(p36p34),+8, t(9;15)(p22;q24)[19]/ 46,XY[1]	ish t(9;15)(JAK2-;JAK2+)[20].nuc ish(JAK2×2)[200],(KMT2A×2)[200], (CBFB×2)[199/200]		48,XY,del(1)(p36.32p35.1), +del(1)(p36.32p35.1),+8, t(9;15)(p21.3;q23)
27	AML	45,X,-Y,der(8)ins(8) (p23q21.3q24)t(8;21) (q21.3;q22), der(12)t(8;12)(q24;q13), der(21)t(12;21)(q13;q22)[20]	ish der(8)ins(8)(D8S504+, RUNX1T1+) t(8;21)(RUNX1T1+,RUNX1+),der(12) t(8;12)(VIJyRM2053+),der(21)t(8;21) (RUNX1+, RUNX1T1+)[5].nuc ish (KMT2A×2)[200],(CBFB×2)[200]	*RUNX1:* *:RUNX1T1*	45,X,-Y, der(8)ins(8)(p21.3q21.3q24.13) t(8;21)(q21.3;q22.12), der(12)t(8;12)(q24.13;q13.13), der(21)t(12;21)(q13.13;q22.12)
15	AML	46,XX,dic(5;10)(q12;q21),+21[18]/52,sl,+1,+2,+9,+11,+20, +21[2]	ish dic(5;10)(wcp5+,D5S23+,EGR1-; wcp10+)[19]		46,XX,der(5)t(5;10) (q11.2q12.3;q11.21q21.1), dic(5;10)(q13.3;q11.21),+21
02	AML	44~47,XX,del(5)(q1?3q3?3), der(7)t(7;17)(p14;q12), i(8)(q10),del(9)(p1?3p1?5), del(12)(p11),-15,-16,-17, +mar[20]	ish del(5)(EGR1-)[20],der(7)(TP53-, D7Z1+,D7S486+) [20].nuc ish (D5S23×2,EGR1×1)[95/100],(D7Z1, D7S486)×2[200],(ETV6×1)[100],(TP53×1, D17Z1×2)[100]		45,XX,del(5)(q14.3q34), der(7)t(7;17)(p22.3;q12) del(7)(p21.3p15.2)del(7)(p14p11.2), idic(8)(p11.21),del(12)(p11.21),del(15) (q11.2q22.2),del(16)(q11.2),-17
04	AML	46,XY,der(4)t(4;13) (q21;q3?3),t(7;14)(q21;q32), add(8)(p21), der(13)inv(13)(p11q?13) t(4;13)(q21;q21)[12]/ 46,XY[3]	ish der(4)t(4;13)(wcp4+,FIP1L1+, CHIC2+,wcp13+)[7],t(7;14)(wcp7+, D7Z1+; wcp7+,D7S486+)[7],add(8) (wcp8+,FGFR1+)[7],der(13)inv(13) (wcp13+)t(4;13)(wcp4+)[7].nuc ish (FIP1L1,CHIC2,PDGFRA)×2 [200],(FGFR1×2)[200], (KMT2A×2)[200],(CBFB×2)[200]		46,XY,t(3;17)(p22.1;p13.1), der(4)t(4;13)(q22.1;q34), del(6)(q14.1q14.2),t(7;14)(q21.2;q32.2),er(8)t(8;17)(p21.2;q21.2), der(13)del(13)(q14q31.3)t(4;13), der(13)t(13;13)(q22.1;q31.3)

^1^ according to the ISCN 2020. ^2^ karyotype formula written according to OGM results: only SV or CNV involving chromosomal segments of ≥10 Mb were considered to be detectable on the karyotype. * CBA performed on the concomitant blood sample with 74% blasts showed a normal karyotype in 20 metaphases. Colors in columns 2 and 6 indicate the genetic group according to the number of abnormalities on standard testing and the discrepancies between OGM versus standard testing as described in Figure 1.

**Table 3 cancers-15-02131-t003:** Detailed results of standard testing and OGM for 5 cases with additional information provided by OGM.

#ID	Leukemia Type	Standard Karyotype ^1^	FISH	Fusion Transcripts	Karyotype Description Based on OGM ^2^	Cryptic Clinically-Relevant Aberrations Detected by OGM ^3^		
						SV Type	Region (Coordinates)	Genes of Interest
03	B-ALL	49,XX,+6,+8, t(9;22)(q34;q11), t(12;13)(q12;q?21), +der(22)t(9;22)[8]/ 46,XX[12]	ish der(12)t(12;13)(ETV6+)[20]. nuc ish(KMT2A×2)[200], (ETV6,RUNX1)×2[200], (TCF3×2)[200]	*BCR:* *:ABL1*	49,XX,+6,+8,t(9;22)(q34;q11), t(12;13)(q12;q14.3), +der(22)t(9;22)[8]/46,XX[12]	deletion deletion deletion deletion deletion deletion deletion translocation interchr	7p12.2(50324504_50399656) 9p21.3(21831433_21996139) 9p21.3(21874513_22488706) 9p13.2(37008719_37031741) 10q25.1q25.2(110008672_110113010) 12p12.1(25245614_25388171) 12q21.33(91882647_92145130) t(12;13)(q12;q14.3)	*IKZF1* *MTAP, CDKN2A* *CDKN2A, CDKN2B* *PAX5* *ADD3* *KRAS* *BTG1* *ARID2;WDFY2*
05	T-ALL	46,XX[20]	nuc ish(TLX3×2)[200],(ABL1×1, BCR×2)[105/200],(KMT2A×2) [200],(TLX1×2)[200]		46,XX	bi-allelic deletion deletion deletion deletion	9p21.3(21903070_23549634) 9q34.11q34.13(128649360_131166096) 12p13.2p12.3(10511129_15366319) 12p13.2p13.1(11475434_12738020)	*CDKN2A,CDKN2B* *SET::NUP214* *ETV6,CDKN1B* *ETV6,CDKN1B*
11	T-ALL	46,XY[22]	nuc ish(TLX3×2)[200],(ABL1, BCR)×2[198/200],(TLX1×2)[200], (KMT2A×2)[196/200]		46,XY,del(6)(q13q22.1)	deletion deletion deletion	9p22.1p21.3(19614301_22695036) 9p21.3(20527876_23828989) 14q11.2(22234858_22563632)	*CDKN2A,CDKN2B* *CDKN2A,CDKN2B* *TCRA/D locus*
29	T-ALL	46,Y,t(X;10)(p11;p12), t(1;7)(p36;q12),der(19) ins(19;7)(p13;q3?4q22)[13]/46,XY[4]	ish t(X;10)(wcpX+;wcpX+, TLX1+)[10],t(1;7)(D7S486+; D7Z1+)[19],der(19)ins(19;7) (D7S486+)[19].nuc ish(TLX3×2) [200],(D7Z1×2,D7S486×3) [182/200],(TLX1×2)[200]		46,Y,t(X;10)(p11;p12),t(1;13;7) (p36.32;q32.3;q11.21), dup(7)(q22.27q36.3)	translocation interchr translocation interchr	t(1;7)(p36.32;q11.21) t(1;13)(p36.32;q32.3)	*TP73* *TP73*
01	AML	47,XX,+mar[17]/46,XX[3]	ish der(?8)(?cen::8q24.?2→ 8q22::8q22→8qter) (wcp8+,RUNX1T1-,MYC++)[18]		47,XX,+der(?)(?cen::8q24.3→ 8q22.1::8q22.1→8qter)	duplication	11q23.3(118461867-118479068)	*KMT2A*

^1^ according to the ISCN 2020. ^2^ karyotype formula written according to OGM results: only SV or CNV involving chromosomal segments of ≥10 Mb were considered to be detectable on the karyotype. ^3^ Chromosomal positions are indicated in GRCh38/hg38 reference genome. Only selected genes of clinical interest are reported. Interchr = inter-chromosomal. Colors in columns 2 and 6 indicate the genetic group according to the number of abnormalities on standard testing and the discrepancies between OGM versus standard testing as described in Figure 1.

## Data Availability

Due to institutional regulation, patient personal data cannot be made publicly available. However, we can respond to acceptable requests that can be sent to the corresponding authors.

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
