# Peer review of "Optical Genome Mapping in Routine Cytogenetic Diagnosis of Acute Leukemia"

_cancers, 2023, doi:10.3390/cancers15072131_

Round 1

Reviewer 1 Report

The authors' purpose was to compare OGM technique results to standard of care karyotyping+FISH+RT-PCR in a cohort of AML and ALL, described in a well-written manuscript.  Below some specific comments : 

-WHO and ELN are not only cytogenetic classifications and further genetic testing are required to properly classify AML patients. At least, the authors should provide informations about NPM1, FLT3-ITD, bZIP CEBPA and TP53 mutation status for AML patients without any defining abnormality.

-Patient #6, the authors did not check/discuss the loss of Y chromosome subclone detected only by OGM. Is it belonging to the inv(16) clone ?

-Patient #7, t(5;5) was supposed to be a paracentric inversion as no deletion or translocation of EGR1 were detected by FISH but figure S4 (although truncated) suggests that its orientation is +/+, rejecting the hypothesis. As t(8;12) and t(13;13), this anomaly was not retained by the authors. However they should check if they are still detected by the high coverage de novo pipeline. 

-Patient #29, The authors should also check if de novo pipeline is able to correctly identify ins(19;7).

-Patient #01, the authors should precise if the KMT2A-PTD was confirmed elsewhere (sequencing?).

Then some minors corrections : 

-minutes chromosomes should be written as "dmin" as required  according to ISCN

-Table 1, patient #6, diagnosis "CBFB::MYH11" is incorrectly written

-Table 2, the extra space in the fusion transcripts "PML: :RARA" must be deleted

Reviewer 2 Report

The paper contains very detailed and implemented results and is a valuable contribution to research conducted in this topic.

Author Response

We thank the reviewer for a favorable review of our manuscript.

Reviewer 3 Report

The authors present a series of patients with acute leukemia on whom they performed optical genome mapping in addition to conventional cytogenetic analysis, FISH, and RT-PCR. They conclude that optical genome mapping could be a reasonable alternative to standard testing by providing a streamlined approach to diagnosis with the caveat that cost savings would currently be limited due to the high price of reagents and equipment.

The manuscript is well-written though there are a few opportunities to make things clearer to the reader. In the introduction, lines 53-54, the authors could define what they mean by "cytogenetic alterations," whether this would extend to molecular genetic abnormalities or solely those detected by chromosome analysis and targeted FISH and RT-PCR assays. In line 157, they should state explicitly whether the sex ratio of 1.31 was male:female or female:male. In this same paragraph, they should explain why there were 29 samples but Table 1 includes samples going up to case 31--stated another way, there are no results in this table for cases 11 and 14. This should actually be changed throughout the text, tables, and figures because it makes little sense to skip two cases, which I found extremely confusing--the authors need to renumber the cases. A weakness of the cases is the skewing toward AML cases, which represented nearly 83% (24/29) of their cases--there were only 3 B-ALLs and 2 T-ALLs so any conclusions made about B-ALL and T-ALL need to be interpreted cautiously.

Figure 1 in the blue box in the legend should read "or" instead of "ou."

Please double-check FISH results for accuracy/proper use of ISCN (nomenclature).

The title for Table 3 could modified to state "... OGM for 5 cases with additional...." Table 2 only contains 24 cases when I was expecting all 29 cases.

The case numbers with only single digits are introduced in Table 1 as #ID 1, 2, 3, etc., but mostly listed in the text as 01, 02, 03. Please keep numbering consistent throughout the text and tables/figures--that is, choose one way or the other.

In lines 246-247, it is speculated that karyotyping for case 18 was "normal probably because of an unsuccessful cell culture." This may be true, however, the t(12;21) that leads to ETV6::RUNX1 is cytogenetically cryptic and this needs to be mentioned in this sentence (I thought that something like 80% of cases with B-ALL with t(12;21) are accompanied by other structural or numerical abnormalities.)

Please place arrows in the karyograms pointing to the abnormal chromosomes in Figures 2 and 3 and in the supplementary materials.

I'm not sure the statement in lines 396-399 about "pediatric cases" is accurate. For both acute and chronic leukemias and both pediatric and adult cases, the leukemic burden is similar at about 10^12 tumor cells, and the amount of hematopoietically active ("red") marrow normally present in children and adults is very similar. If the authors can find a reference to support the inference that children with leukemia might have lower yields of leukemic cells for analysis, then the reference should be provided. Otherwise, it seems to be an unsupported statement favoring optical genome mapping over the standard of care.

Finally, in the supplementary materials, in addition to adding arrows to the abnormalities in the karyograms, the authors should enlarge the size of the circos plots--and also the visualization of the deleted region on the UCSC genome browser in Figure S6 B. If the images are so small and difficult to read, there is almost no reason to include them--they should be large enough that the reader can see the details. The legend for S5, line 37, should read "CNV" instead of "CVN."

Overall, this is a very nice study, and these comments and suggestions are relatively minor and easy to remedy.
